# Overview of Current Targeted Anti-Cancer Drugs for Therapy in Onco-Hematology

**DOI:** 10.3390/medicina55080414

**Published:** 2019-07-28

**Authors:** Stefania Crisci, Filomena Amitrano, Mariangela Saggese, Tommaso Muto, Sabrina Sarno, Sara Mele, Pasquale Vitale, Giuseppina Ronga, Massimiliano Berretta, Raffaele Di Francia

**Affiliations:** 1Hematology-Oncology and Stem Cell Transplantation Unit, Istituto Nazionale Tumori, Fondazione “G. Pascale” IRCCS, 80131 Naples, Italy; 2Gruppo Oncologico Ricercatori Italiano GORI ONLUS, 33100 Pordenone, Italy; 3Hematology and Cellular Immunology (Clinical Biochemistry) A.O. dei Colli Monaldi Hospital, 80131 Naples, Italy; 4Anatomia Patologica, Istituto Nazionale Tumori, Fondazione “G. Pascale” IRCCS, 80131 Naples, Italy; 5Department of Medical Oncology, CRO National Cancer Institute, 33081 Aviano (PN), Italy; 6Italian Association of Pharmacogenomics and Molecular Diagnostics (IAPharmagen), 60125 Ancona, Italy

**Keywords:** anticancer mAbs, Tyrosine kinase inhibitors, tailored therapy, personalized medicine, pharmacogenomics

## Abstract

The upgraded knowledge of tumor biology and microenviroment provides information on differences in neoplastic and normal cells. Thus, the need to target these differences led to the development of novel molecules (targeted therapy) active against the neoplastic cells’ inner workings. There are several types of targeted agents, including Small Molecules Inhibitors (SMIs), monoclonal antibodies (mAbs), interfering RNA (iRNA) molecules and microRNA. In the clinical practice, these new medicines generate a multilayered step in pharmacokinetics (PK), which encompasses a broad individual PK variability, and unpredictable outcomes according to the pharmacogenetics (PG) profile of the patient (e.g., cytochrome P450 enzyme), and to patient characteristics such as adherence to treatment and environmental factors. This review focuses on the use of targeted agents in-human phase I/II/III clinical trials in cancer-hematology. Thus, it outlines the up-to-date anticancer drugs suitable for targeted therapies and the most recent finding in pharmacogenomics related to drug response. Besides, a summary assessment of the genotyping costs has been discussed. Targeted therapy seems to be an effective and less toxic therapeutic approach in onco-hematology. The identification of individual PG profile should be a new resource for oncologists to make treatment decisions for the patients to minimize the toxicity and or inefficacy of therapy. This could allow the clinicians to evaluate benefits and restrictions, regarding costs and applicability, of the most suitable pharmacological approach for performing a tailor-made therapy.

## 1. Introduction 

Targeted therapies are developed to encompass the nonspecific toxicity associated to standard chemo-drugs and also to ameliorate the efficacy of treatment. Biological agents can be used alone, but very often a combination of targeted molecules and conventional anti-tumor drugs is used. This new strategy aims to a selectively killing of malignancy cells by targeting either the expression of specific molecules on cancer cell surface or the different activated molecular pathways that directed to tumor transformation [1]. New therapeutic approaches include a combination of “old” anticancer drugs (i.e., chemotherapies) and innovative molecules (targeted agents). These procedures are planned to mark both primary and metastatic cancer cells. The current classification of cancer-hematology targeted drugs includes monoclonal antibodies (mAbs), small-molecule inhibitors (SMIs), interfering RNA (iRNA), microRNA, and oncolytic viruses (OV). Their mechanisms of action can be either tumor specific (by interfering with cancer cell membrane biomarkers, cell-signaling pathways, and DNA or epigenetic targets), or systemic, through triggering of the immune responses [2,3,4].

Currently, scientific evidence is still low, to address therapeutic drug monitoring of mAbs and SMIs. In this scenario, their combination with adjuvant therapies may represent a promising cancer treatment approach [5]. 

In 2001, the first selective ABL Tyrosin Kinase inhibitor (TKI, Imatinib) was approved by the US Food and Drug Administration (FDA) [6]. This was quickly followed by the monoclonal antibody marking the CD20 antigen (anti-CD20mAb, Rituximab) [7]. Both agents accounted the revolution in the management of patients with Chronic Myeloid Leukemia (CML) and non-Hodgkin’s lymphoma (NHL), respectively. These drugs guarantee around 80% response rate; on the other hand, drug-resistance is still a limitation, and new generation TKIs are being developed to by-pass these matters. 

Currently, various new molecules are developed against specific tumor cell targets. Among these, histone deacetylase inhibitors (HDACi) and DNA hypomethylating agents target genetic/epigenetic determinants central for tumor growth ablation. New peptide vaccines targeting novel tumor-associated antigens, alternative checkpoint inhibitors, and chimeric antigen against T receptor (CAR-T) of the lymphocytes are being developed for the patients who have failed classical immunotherapy (often anti-PD1/anti-PD-L1). Newer SMIs targeting a variety of oncogenic signaling are being advanced to overcome the emerging of resistance phenomena to the existing targeted drug procedures [8]. Above Table 1 reviews currently open phase I/II/III clinical trials for onco-hematologic patients. The data in the table were collected from clinicaltrils.gov (accessed on June 2019). It may provide a useful implement to oncologist who treat relapsed refractory hematology malignancy.

Thanks to unique mechanisms of action these drugs are part of the therapy for many anticancer protocols, including lung, breast, pancreatic, and colorectal tumors, as well as lymphomas, leukemia, and multiple myelomas (MM). Also, targeted therapies call for new approaches to evaluating treatment effectiveness, and to assess the patients’ adherence to treatment [9].

Although targeted drugs, primarily the mAbs administration, are better tolerated than conventional chemotherapy, they are recorded numerous adverse events, such as cardiac dysfunction, hypertension, acneiform rash, thrombosis, and proteinuria [10]. Since, SMIs are metabolized preferentially by cytochrome P450 enzymes (CYP450), the pharmacokinetics variation and multiple drug interactions should be present [11].

Finally, targeted therapy has raised new questions about the tailoring of cancer treatments to each patient’s tumor profile, the estimation of drug response, and the public reimbursement of cancer care. In agreement with these and other criticisms, the intention of this review is to provide information about the overall biology and mechanism of action of target drugs for onco-hematology, including overall pharmacodynamics due toxicities, and eventually inefficacy of the therapy [12]. The well-noted correlation between the individual genomics profile and overall pharmacokinetic are reported below. In addition, the economic impact of these drugs, which can exceed several-fold the cost of traditional approaches, may become a major issue in pharmacoeconomics; an early evaluation of outcomes and challenges are also reported in this issue. 

## 2. The Biological Mechanism of Action of the Targeting Agents

Traditional cytotoxic chemotherapy works mainly through the inhibition of cell division like as cytoskeleton inhibitors (i.e., tecans, vincristine)analogues of nucletides (i.e., 5-Fluorouracil, Gemcitabine). So, each normal cell in mitotic stage (e.g., hair, gastrointestinal epithelium, bone marrow) is affected by these drugs. In contrast, the mechanism of action of targeted agents block specifically the proliferation of neoplastic cells by interfering with specific pathways essential for tumor progression and growth (Figure 1). It is true that some of these molecular targets could be present also in normal tissues, but they are often abnormally expressed in cancer tissues because of genic alterations.

### 2.1. Monoclonal Antibodies 

In the last decade, about 30 mAbs have been approved, almost the 50% of them for the hematology-cancer therapy. The backbone of a mAb includes the fragment antigen binding (Fab), which recognizes and engage antigens, thus targeting the specific counterpart on cancer cells which is the objective of such therapies [13]. These antibodies agent, either as native molecules or conjugated to radioisotopes or toxins (viral molecules to interfere with cell vitality) exert their antineoplastic effects through different mechanisms: by engaging and arming host immune effectors (i.e., natural killer cells and the complement) against target cells; by binding to receptors or ligands, thereby interfering with essential cancer cell proliferation pathways; or by carrying a “lethal cargo”, such as a radioisotope or toxin, to the target cell [14,15]. Because they are protein in structure, mAbs denatured in the gastrointestinal tract, and for this reason, they are administered intravenously. In this way, they do not undergo hepatic metabolism thus escaping significant drug interactions via CYP450 pathway.

Latest 30 years, as biotechnology has evolved, the design of mAbs has changed. Immunized mice developed early molecules in this class against target antigens. The resulting mAbs comprised the entire murine immunoglobulins (Igs), which were, hypothetically, carrying a risk of antigenic reaction during parenteral administration. The primary adverse events for patients treated with these new drugs is forming anti-mouse immunoglobulin antibodies, which could counteract the effect of the therapeutic mAbs. To decrease these adverse reactions, was developed mAbs contain an enlarged proportion of human Ig components to reducing murine Ig components. So mAbs are defines as: chimeric antibodies (about 65% human), humanized antibodies (95% human), and human antibodies (100% human) [16].

General precautions of mAbs include: (a) Co-administrations of vaccines may be avoided; (b) Herpes and Pneumocystis prophylaxis recommended in therapy influencing the immune surveillance system (i.e., Anti-CD52);

The mAbs were designed to affect several cellular pathways. An angiogenic inhibitor (i.e., Anti-VEGF) was currently investigated in NHL, MM, and myeloproliferative disorders [17]. The major side effect regardless the risk of ischemic events.

Immunomodulators aimed to achieve blocking of inhibitory molecules on the cell surface of immune effectors (i.e., Anti PD-1, anti-CD30, anti CD52, anti-CTLA-4, and anti-CD80), were approved for the treatment of patients with Hodgkin Lymphoma (HL). 

The first targeted therapies have developed through the study of the over/expression of markers on the surface of tumor cells (on lymphoma and leukemia cells) including the cluster of differentiation 20, 33, and 52 (CD20, CD33, and CD52). Targeting this molecule affects the overall humoral immune response, as CD20 is also expressed on normal lymphocyte B. This estimation has led to the administration the anti-CD20 mAb Rituximab for the treatment of autoimmune diseases such as rheumatoid arthritis [18], and NHL [19].

Moreover, mAbs might induce tumor cell death through direct and indirect mechanisms, all prominent in blocking several and different oncogenic pathways. mAbs are effective at multiple levels: surface molecules (neutralizing ligand-specific receptor interaction), receptors (competitive with a ligand for binding); receptor signaling (signal blocking), or intracellular molecules. Therapeutic mAbs, which are typically water soluble and large molecular weight (about 160 kDa), target extracellular molecule-inducing signals, such as receptor-binding sites and ligands. 

Noteworthy, Brentuximab vedotin have a new peculiar mechanism of action. It is an anti-CD30 antibody-drug conjugate (mAbC) that delivers an anti-tubulin toxin that induces apoptotic cell death in CD30-expressing tumor cells. Brentuximab vedotin is a mAb with a regular action involving a multi-step progression: binding to CD30 on the cell surface recruits internalization of the mAbC-CD30 complex, which then transfers to the cytoskeleton structures, where the released toxin, the monoauristatin-E, rescinds the microtubule complex, inducing cell cycle arrest and apoptotic death of the CD30-expressing cell in replication stage. However, classical Hodgkin’s lymphoma (HL) and T-anaplastic large cell lymphoma (ALCL) express CD30 as an antigen on the surface of their malignant cells [20].

### 2.2. Small Molecule Inhibitors

SMIs are chemical agents with low molecular weight (inferior to 1 kDa) capable of entering into the targeted cells, thus blocking signaling pathway and interfering with downstream intracellular protein systems (i.e., TK signaling). These molecules consist of TKI, BKI, Aurora Kinase Inhibitors (AKI), proteosome Inhibitors and HDAC inhibitors. Tyrosine Kinases phosphorylate specific amino acid residues of intracellular substrates affecting angiogenesis and cell growth in normal and malignant tissues.

The SMls differ from mAbs in many ways: (i) they are usually orally administered rather than parenterally infused; (ii) they are generated via a step-by-step chemical practice, a method that is often much less expensive than the bioengineering (i.e., recombinant DNA techniques) required for mAbs generation [18]; (iii) they reach fewer definite targeting than mAbs, as marked in the multitargeting nature of the kinase inhibitors [21].

In contrast to mAbs, most of the SMIs are metabolized by the CYP450 enzymes family, which may consequence in drug-drug interactions with molecules such as warfarin, azole antimicotics, viral anti-protease, and the St. John’s wort, known as unyielding CYP450-inhibitors [22].

The mAbs have administered once every one-to-four weeks due to pharmacokinetic of T_1/2_ ranging from few days to few weeks. While most SMIs have half-lives of few hours and necessitate daily dosing., with the exception of few molecules (i.e., bortezomib) which is given intravenously, SMIs are administered per os (orally).

#### 2.2.1. Kinase-Based Signaling Inhibitors

Agents developed for blocking the signaling pathway includes: Tyrosin Kinase Inhibitors (TKIs), Bruton Kinase Inhibitors (BTKIs), Aurora Kinase Inhibitors (AKI) and SYK, PI3K Inhibitors.

Imatinib is the first FDA-approved SMIs (since 2002) for the treatment of CML. It inhibits a constitutive TK activities resulting from the translocation of chromosomes 9 and 22 (the Philadelphia chromosome), involving BCR and Abl genes. Because this molecular anomaly occurs in essentially all patients with CML, imatinib therapy results in a complete hematologic response in 98% of them [23]. Novel TKI molecules to overcome the imatinib resistance due to ABL T315I was developed, Bosutinib [24], Dasatinib [25], Nilotinib [26], and Ponatinib [27]. Furthermore, these small molecules are not able to prevent cerebral metastasis [28].

Other altered TK pathway, involved in myeloid and lymphoproliferative disorders are the Janus Kinase receptors (JAK1, JAK2, and JAK3), phosphatidylinositol-3-kinase (PI3K), MET mitogen-activated protein kinases (MAPK), RET, and RAF. 

JAK family (JAK1, JAK2, and JAK3) members mediate cytokine signaling via downstream activation of the STAT family of transcriptional regulators. Activated STATs endorse multiple cellular actions, like differentiation, proliferation, migration, and apoptosis. In particular, JAK2 is mediated hematopoiesis via GM-CSF, leptin, IL3, erythropoietin, and thrombopoietin [29]. Activation of JAKs are induced conformational changes of ligand-binding to cytokine receptors, which engage phosphotyrosine-binding domains and/or SRC homology-2, which leads to activation of STAT, Ras–MAPK, and PI3K–AKT signaling pathways. Mutations in JAK2 exon 12 in pseudokinase domain (JH2) and JAK2 Val617Phe in exon 14 have been identified in approximately 4% and 90% of PV cases respectively. Several agents to interfere with JAK/STAT pathway were deigned: Itacitinib is a selective JAK1 inhibitor [30]; Momelotinib and Ruxolitinib are both JAK1-2 inhibitor [31,32]. 

MET, is a receptor tyrosine kinase that, after binding with hepatocyte growth factor (HGF), activates the PI3K and MAPK pathways. Over-expression of MET gene and exon-14-skipping mutations are characteristic abnormalities causing increased MET signaling progression. Cabozantinib is unique MET Inhibitor studied in phase 1b study in patients with relapsed MM [33]. 

The PI3K pathway is an essential mediator of cell survival signals. Several molecules PI3K-p110 α,γ,δ sub unit inhibitor were under trial phase 1–2 for NHL, like as AMG319 [34], Buparlisib [35,36], Copanlisib [37], CUDC-907 with additional inhibition of HDAC [38], Dactolisib with additional inhibition on mTOR-p70S6K [39], Duvelisib [40], Getadolisib is a pan-PI3K, mTOR inhibitor. Its safety and maximum tolerated dose have been recently established in a phase 2 study of AML and CML [41]. Idelalisib currently approved use for CLL [42,43], INCB040093 [44], Pictilisib [45], and Taselisib the ultra-selective isoform-sparring PI3K-p110 α,γ,δ [46], TGR-1202 [47] also tested in combination with brentuximab in HL [48] and Voxtalisib [49].

The RET proto-oncogene encodes a receptor TK for members of the glial cell line-derived neurotrophic factor (GDNF) family. Vandetanib [50], sorafenib [51,52], sunitinib [53], and cabozantinib are multi-targeted TKIs with RET-blocking activity. They are currently tested in phase 2 trials for MM and ALL [54]. 

BRAF is a downstream signaling mediator of KRAS, which activates the MAP kinase pathway. BRAF mutations are found in about 99% of cases of hairy cell leukemia [55] and in several cases of MM. To date, Dabrafenib and Vemurafenib are FDA-approved for BRAF V600E-positive malignant melanomas, but clinical trials of phase 1–2 were currently performed [55]. 

#### 2.2.2. Bruton’s Tyrosin Kinase Inhibitor

BTK is a cytoplasmic TK essential for B-lymphocyte development, differentiation, and signaling. As known, humans X-linked agammaglobulinemia (XLA) caused by mutations in the BTK gene (usually D43R and E41K). Activation of BTK triggers step by step signaling events that culminates in the cytoskeletal rearrangements by Ca^2+^ recruitments and transcriptional regulation involving nuclear factor-κB (NF-κB). In B lymphocytes, NF-κB bind the BTK 5′UTR gene promoter and active gene transcription, whereas the B-cell receptor-dependent NF-κB signaling pathway requires functional BTK [56,57]. 

Based on these issues several molecules were developed to target BTK: Ibrutinib [58], and idelalisib [42].

#### 2.2.3. Histone Deacetylase (HDAC) Inhibitors 

Gene expression is regulated primarily by histone acetylation and deacetylation mechanism; this process is essential for relaxing the condensed chromatin and exposes the promoter regions of genes to transcription factors. On the contrary, deacetylation catalyzed by histone deacetylases (HDACs) results in gene silencing [59]. In leukemic cells, this equilibrium is disturbed, and therefore, HDAC inhibitors emerged as a striking therapeutic approach to revise therapy. Unlike, clinical activity of monotherapy with an HDAC inhibitor was low, with ORR of 17% for vorinostat and 13% for mocetinostat [60], and no clinical response was achieved with entinostat monotherapy [61]. In addition, Belinostat has been tested in elderly patients with relapsed AML [62]. Since it is primarily metabolized by UGT1A1; the initial dose should be reduced if the beneficiary is acknowledged to be homozygous for the UGT1A1*28 allele [63]. Therefore, current studies focus on combination regimens of HDAC inhibitors with other epigenetic agents are still ongoing.

#### 2.2.4. Proteosome Inhibitors

The proteasome is the latest promising molecular target for cancer therapy, a large multimeric [64]. The proteasome is a protein complex that degrades the damaged proteins, and neoplastic cells are highly needy on increased protein production and degradation. It has a crucial role in cell signaling, cell survival, and cell-cycle progression. For these issues, proteasome inhibition is a core of therapy in lymphoid malignancies. Furthermore, proteasome inhibitors, such as bortezomib [65] and carfilzomib [66], are currently integrated into major regimens for multiple myeloma (MM) and mantle cell lymphoma (MCL) patients. Recently, proteasome inhibitors have also been used for the treatment of the relapsed/refractory setting for other NHLs, such as follicular cell lymphoma (FCL) [67,68]. 

## 3. Pharmacogenetics, Overall Pharmacokinetics

Genetic tests for targeted cancer therapy detect either acquired mutations in the DNA of cancer cells and or polymorphisms in all germinal cells. Genotyping the cancer cells can help guide the type of targeted treatment, and it can predict who may respond to planned therapy and who is not likely to have benefited.

Researchers have extensively studied these variants in genes in order to better understand cancer and to develop drugs to interfere with a specific step in cancer growth while doing minimal damage to normal cells. The first example was Dasatinib and Nilotinib designed to by-pass the acquired mutation T315I in *Abl* gene in LMC patients. Unluckily, not every cancer has these acquired mutations, and various hematologic-cancers cells without this genetic signature cannot benefit to personalized treatments.

Pharmacogenomic tests are important, also, for the pharmacoeconomic issue: targeted drugs are expensive, and they generally work efficacy only in cancer patients who carried a genetic marker. Genetic testing prior to beginning therapy is necessary to match the treatment up with the patients and cancers likely to benefit from them. In contrast, chemotherapic drugs are cheaper, but it based on the paradigm of the “trial and error” leading the hospitalization during treatments. Recently, to reduce pharmaceutical expenditure, a combination of 2 SMIs blocking 2 mutated genes (*BRAF* and *MET*) are performed in the unique formulation (encorafenib + binimatenib) for melanoma therapy. So, the route is also drawn for the onco-hematology [137].

The most common targeted cancer drugs for which tests are available include:

Drugs that block growth signaling binding to receptors on the cell surface

Small molecules inhibitors are able to cross the cell membrane and block downstream the growing signals in the specific active site.

These pharmacogenetic tests are used to help adjust drug dosage for certain cancers. They help to inform the oncologist as to whether certain targeted cancer drugs may or may not work.

## 4. Outcomes and Challenges

Generally, chemotherapy is administered intravenously in an ensured infusion area. Therefore, patient adherence to treatment regimens is gladly reviewed. Delays and omissions in chemotherapy dosages, whether they are the result of patient preference or treatment-related either toxicities or resistance, are immediately recognized and documented. In contrast, most SMIs (almost all are oral formulation) administered at home on a long-term daily scheduling, could be difficult to recognize adverse events in real time. Thus, the task of assessing patient adherence more closely resembles that encountered with therapies for chronic diseases such as diabetes and hypercholesterolemia, and in so-called “frail patients”. Finally, a few studies published to date shown high variability and unpredictability about patient adherence to oral cancer treatment regimens [138].

Targeted therapy offers new means to determine the best possible treatment. Moreover, estimation of treatment success could not be based on the reduction in neoplastic volume and/or evaluation of toxicity through the degree of myelosuppression severity as well as traditional chemotherapy. Targeted therapies could convey a clinical benefit by stabilizing tumors, rather than reducing the progression of the neoplastic population cells. It also must necessitate to a paradigm shift in the evaluating the effectiveness of therapy. To set up the optimum of targeted drugs in terms of dosing and efficacy, must be evaluated progressively several endpoints, such as tumor metabolic activity on positron emission tomography (PET) scans, levels of circulating neoplastic cells, and following levels of target molecules in tumor tissue [139]. These actions introduce complexity and cost to medical activity. Also, repeated biopsies of tumor tissue may be untimely for patients and improper to institutional evaluation boards.

Even though these procedures introduces new economic considerations: (i) oral SMIs eliminate treatment costs associated with the hospitalized intravenous infusions; (ii) the biotechnical production of mAbs can improves costs exponentially; (iii) need to genotype or phenotype tumor tissue; and (iv) new competencies for oncologist about pharmacogenomic testing [140].

In term of the expenses, the targeted therapy is widely most expensive than traditional approaches. For example, in colon cancer, multidrug treatment regimens containing bevacizumab or cetuximab increase the cost to about 500-fold ($30,790 for 8 weeks of treatment), compared with fluorouracil/leucovorin-based regimen ($63 for the same period) [141].

Based on this consideration, clinicians and laboratorists have a duty to cooperate in evaluating the advantages and limitations, particularly regarding costs and applicability, of the pharmacogenomic tests most appropriate for routine incorporation in clinical practice [142].

## 5. Conclusions and Future Outlook

For decades, the hallmark of anticancer treatments has based on the cytotoxic chemotherapy. These molecules rapidly target mitotic cells, including, unluckily, not only cancer cells but also, normal tissues in physiological growing phase. As a result, many patients develop the general adverse drug reaction (ADR) toxicities such as gastrointestinal symptoms, myelosuppression, and alopecia.

The medical approach based on chemotherapy alone, or in combination with surgical, and/or radiation, has reached doubtful therapeutic advances in spite of many significant enhancements in such treatments. While chemotherapy rests the primarily support of the current treatment in onco-hematology, it is limited by a narrow therapeutic index, noteworthy toxicity leading to treatment discontinuations and frequently acquired resistance.

The targeted molecules are drugs manufactured to interfere with specific proteins necessary for tumor growth and progression.

The new oncologic challenges of the 3rd millennium are based on the developments of the targeted therapy, including OV, interfering RNA molecules, and microRNA [143].

The OV is capable of selectively replicating in tumor cells, leading to their lysis. There is growing proof, in preclinical studies, on the combining synergistic effects of OV to several chemotherapies [144].

However, it is imperative to define the molecular mechanisms involved in the beneficial aspect of an individual therapy approach, despite the medicine adopted. This field is now translated in the clinical application for Chimeric Antigen Receptor (CAR) T-Cell Therapy. Moreover, microRNAs, a family of small noncoding RNAs implicated in the anti-cancer activity of many therapeutic agents, are shown to serve as attractive targets for the oncogene c-Myc-based combination therapy [145].

The mechanism of oxidative stress repeatedly described as a co-factor in cancer development could be a mechanism involved by cancer therapies. Many patients to decrease the side effect often need antioxidant supplementation to adjuvant therapy, which (i.e., Glutathione) [146], natural remedies [147], and other complementary and alternative medicines [148].

In this scenario, the genes encoding the familiar hallmarks of cancer must detect growth factor (GF) and GF receptors, apoptosis, multi-drug resistance, neovascularization, and invasiveness must be analyzed deeply. Also, the genetic predisposition related to factors promoting genome instability, inflammation, deregulation of metabolism and the immune system evasion/damaging must identify [149].

The ribosome-inactivating proteins (RIPs) are promising agents extracted from the plant with a complete damaging of the ribosomal activities [150].

Unlikely, no upgrading was seen for primary lymphoma central nervous system (PLCNS) in term of targeted therapy. The use of SMIs in PCNS and as preventing metastasis has failed [28]. Encouraging evidence supporting the addition rituximab to classical alkylating agents is growing, and a recent randomized trial (MATRix regimen) demonstrated a significantly better OS [151].

Based on this rationale, the oncologist should assess advantages and limits, regarding applicability and expenditure of the most fitting pharmacological approach to performing a customized therapy.

Finally, the Pharmacogenomic tests are mandatory for cancer targeted therapy; it can detect the variants that code mutant proteins markers, thus identifying tumors that may be susceptible to targeted therapy.

## Figures and Tables

**Figure 1 medicina-55-00414-f001:**
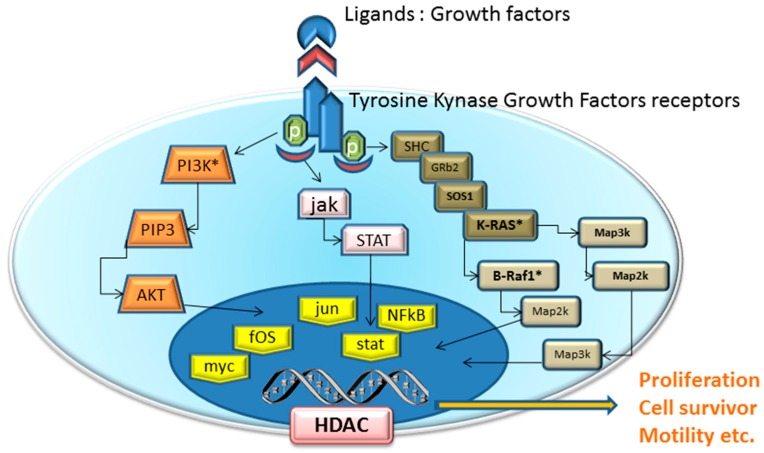
Schematic of pathway inhibition by targeted agents and their effects on cell proliferation, apoptosis, metastasis and angiogenesis. Receptors for growth factors (VEGFR, FGFR, PDGFR) activate intracellular receptor tyrosine kinases (RTKs) and the downstream RAS/RAF/mitogen-activated protein extracellular kinase (MEK)/extracellular signal-regulated kinase (ERK) signaling pathway, and promote the growth, migration, and morphogenesis of vascular endothelial cells, thus increasing vascular permeability.

**Table 1 medicina-55-00414-t001:** The most common agents suitable for onco-hematology targeted therapy.

Agent (Brand Name)	Target(s)	FDA-Approved Indication(s)	Clinical Trials ^$^	Toxicities, Side Effects, and Precautions	Pharmacogenomic Annotations
AFM13 [69]	CD30/CD16A	Under revision	HL—phase I/II. (P)	ND	ND
Alvespimycin [70] (IPI-493; 17-AG)	HSP90	Under revision	MM—phase I. (P)	ND	ND
AMG319 [34]	PI3K-p110δ	Under revision	NHL—phase I. (P) CLL—phase I (P)	ND	ND
Apitolisib [71] (GDC-0980 (RG7422)	PI3K -p110α/β/δ/γ and mTOR	Under revision	NHL—phase I (P)	ND	ND
Avelumab [72] (MSB0010718C)	PD-L1	Metastatic Merkel Cell Carcinoma	AML—phase I/II. (P) HL—phase I. (U) NHL—phase III.	fatigue, fever, colitis	ND
Belinostat [62] (Beleodaq)	HDAC Inhibitor	Relapsed or refractory peripheral T-cell lymphoma (PTCL)	HL—phase I/II. (U) NHL—phase I/II. (U) AML—phase I/II. (U) CML—phase I. (U) cMPD—phase II. (U) ALL—phase I. (U) MM—phase II (U)	Thrombocytopenia, leukopenia, and anemia. Infection, hepatotoxicity, tumor lysis syndrome, embryo-fetal toxicity	Confirm the presence of t(9;22)BCR/ABL fusion gene.
Bendamustine [73] (Treanda)	alkylating drug	CLL	HL—phase I/II. (U) NHL—phase I/II. (U) ALL—phase I/II. (U) MM and PLCNS—phase I/II (U)	Myelosuppression, infections, tumor lysis syndrome, skin reactions	ND
Bevacizumab [74] (Avastin)	VEGF Inhibitor (VEGF ligand)	Metastatic carcinoma of the colon or rectum	NHL—phase I/II. (P) AML—phase I/II. (P) cMPD—phase I/II. (P) CML—phase I/II. (P) MM and PCN- phase II (P).	Gastrointestinal perforation; wound healing complications; hemorrhage; arterial and venous thromboembolism; proteinuria; hypertension. Discontinue use several weeks before elective surgery; do not restart until surgical incision has healed	VEGFA rs3025000 C > T rs3025033 A > G.
Blinatumomab [75] (Blincyto)	Bispecific CD19 directed CD3/T-cell	Relapsed or refractory B-cell precursor ALL Ph-negative	AML Ph+—phase II. (U) NHL—phase II. (U) cMPD—phase II. (U)	Infections	Confirm the presence of t(9;22)BCR/ABL fusion gene.
Bortezomib [65] (Velcade)	26S proteasome	MM, patients who have received at least two prior therapies and have demonstrated disease progression on the last therapy	HL—phase II. (P) NHL—phase I/II. (U) AML—phase I/II. (P) ALL—phase I. (P) CLL—phase I. (P) cMPD—phase I (U).	Peripheral neuropathy; myelosuppression; rash; constipation; diarrhea; edema; fatigue, malaise and weakness, nausea, appetite decreased constipation, peripheral vomiting and anemia	Monosomy 5/7 FLT3-ITD ALK-1 Del(5q).
Bosutinib [24] (Bosulif)	Kinase inhibitor	Chronic, accelerated, or blast phase Ph + CML, resistant or intolerant to prior therapy	ALL Ph-positive—phase I/II. (U)	diarrhea, nausea, thrombocytopenia, vomiting, abdominal pain, rash, anemia, pyrexia, and fatigue.	Confirm the presence of t(9;22)BCR/ABL fusion gene.
Brentuximab vedotin [20] (Adcetris)	CD30	HL after failure of autologous stem cell transplant (ASCT) or after failure of at least two prior multi-agent chemotherapy regimens in patients who are not ASCT candidates. ALCL after failure of at least one prior multi-agent chemotherapy regimen.	NHL—phase II. (U) cMPD—phase 0/II. (P) AML—phase 0/I/II. (P) ALL—phase 0/I/II. (P) PLCNS—phase I/II. (P)	Neutropenia, peripheral sensory neuropathy, fatigue, nausea, anemia, upper respiratory tract infection, diarrhea, pyrexia, rash, thrombocytopenia, cough, and vomiting.	CD30
Buparlisib [35,36] (BKM120) (NVP-BKM120)	PI3K-p110α/β/δ/γ	Under revision.	NHL—phase I/II. (U) CLL—phase I/II. (U) ALL—phase I. (U) AML—phase I. (U)	ND	ND
Cabozantinib [33] (XL184) (Cabometyx)/(Cometriq)	RET, MET and VEGFR2 Kinase inhibitor	Advanced renal cell carcinoma after prior anti-angiogenic therapy/metastatic medullary thyroid cancer.	MM—phase I/II. AML—phase I. ALL—phase I/II.	diarrhea, fatigue, nausea, decreased appetite, palmar-plantar erythrodysesthesia syndrome (PPES), hypertension, vomiting, weight decreased, and constipation.	Acquired mutation on FLT3 ITD, c-Kit D816V.
Carfilzomib [67] (KYPROLIS)	Proteasome Inhibitor	Multiple myeloma after at least two prior therapies including bortezomib and an immunomodulatory agent and have demonstrated disease progression on or within 60 days of completion of the last therapy.	NHL—phase I/II. (U) HL—phase I/II. (U) ALL—phase I. ND AML—phase I. ND CLL—phase I. ND	Fatigue, anemia, nausea, thrombocytopenia, dyspnea, diarrhea, and pyrexia.	ND
Ceritinib [76] (Zykadia)	ALK	ALK-positive, metastatic non-small cell lung cancer (NSCLC) after progression on or intolerant to crizotinib.	NHl—phase I/II ALK+ (U) AML—phase I. (P) ALL—phase I. (P)	diarrhea, nausea, elevated transaminases, vomiting, abdominal pain, fatigue, decreased appetite, and constipation.	ALK gene rearrangement (FISH) or RT-PCR fusion gene.
Cetuximab [77] (Erbitux)	EGFR (HER1/ERBB1)	Metastatic colorectal carcinoma (EGFR positive) refractory to irinotecan- based chemotherapy (as a single agent or with irinotecan).	MM—phase II (P)	Infusion reaction, dermatologic toxicity, interstitial lung disease, fever, sepsis, kidney failure, pulmonary embolus, diarrhea, nausea, abdominal pain, vomiting.	EGFR protein expression positive KRAS codon 12 and 13 mutations negative.
Copanlisib [37] (BAY 80-6946)	PI3K-p110α/δ	Under revision.	NHL—phase I/II. (U)	Hyperglycemia, hypertension, fatigue, diarrhea, neutropenia, anemia, hypertension.	ND
CUDC-907 [38]	PI3K-p110α and HDAC1/2/3/10	Under revision.	NHL—phase I/II (U) MM—phase (P)	ND	MYC alterations.
Dactolisib [39] (BEZ235)	PI3K- p110α/γ/δ/β and mTOR-p70S6K	Under revision.	ALL—phase I. (P) AML—phase I. (P) CML—phase (P).	ND	t(9;22)
Daratumumab [78,79] (Darzalex)	CD38	MM after at least three prior lines of therapy including a proteasome inhibitor (PI) and an immunomodulatory agent or who are double-refractory to a PI agent.	NHL—phase I/II. (U) CLL-phase I/II. (U) AML—phase I/II. (U) ALL—phase I/II. (U)	Infusion reactions, fatigue, nausea, back pain, pyrexia, cough, and upper respiratory tract infection.	ND
Dasatinib [25] (Sprycel)	ABL BCR-ABL, SRC family, c-KIT, PDGFR	Chronic Myeloid Leukemia or Acute Lymphoblastic Leukemia-Ph positive with resistance or intolerance to prior therapy including imatinib.	NHL—phase I/II (P) HL—phase I/II. (P) CLL—phase II. (U) MM and PCN—phase I/II (W). cMPD—phase I/II. (U) AML—phase II. (U)	Rash; pleural effusion; fluid retention; mucositis; myelosuppression; QT interval prolongation.	Confirm the presence of t(9;22)BCR/ABL fusion gene T315I mutation-positive.
Denileukin [80] Diftitox(Ontak)	CD25/IL2RA (diphtheria toxin)	Persistent or recurrent cutaneous T-cell lymphoma CD25+.	MM—phase I (U) cMPD—phase II. (U) NHL—phase II. (P) CLL—phase II. (U)	Fever, management of vascular leak syndrome or dehydration to secondary to gastrointestinal toxicity.	CD25 protein expression (IHC).
Denosumab [81] (Prolia)(Xgeva)	RANKL	Postmenopausal women with osteoporosis at high risk for Fracture.	MM—phase II/III (U) NHL—phase II. (U) HL—phase II. (U)	back pain, pain in extremity, hypercholesterolemia, musculoskeletal pain, and cystitis.	RANKL protein expression (IHC).
Durvalumab [82]	PDL-1	Under revision.	HL—phase I/II. (U) NHL—phase I/II. (U) MM—phase I/II. (U) CLL—phase I/II. (U) AML—phase II. (U) cMPD—phase II.	Diarrhea, colitis, increased lipase, myasthenia gravis, pericardial effusion, neuromuscular disorder.	ND
Duvelisib [40] (IPI-145) (INK1197)	PI3K -p110δ/γ	Under revision.	NHL—phase I/II/III. (U) CLL—phase I/II/III. (U) ALL—phase II. (U)	ND	ND
Elotuzumab [83]	SLAMF7	Multiple myeloma in combination with lenalidomide and after one to three prior therapies.	LS—phase III. (U)	Fatigue, diarrhea, pyrexia, constipation, cough, peripheral neuropathy, nasopharyngitis, upper respiratory tract infection, decreased appetite, pneumonia.	ND
Entinostat [61] (MS-275)	HDAC Inhibitor	Breakthrough Therapy for the treatment of locally recurrent or metastatic estrogen receptor (ER)-positive breast cancer.	HL—phase II. (P) NHL—phase I. (P) cMPD—phase I/II. (P) ALL—phase I/II. (P) ALAL—phase I. (P) CML—phase I. MM and PCN—phase I.	ND	ND
Entospletinib [84] (GS-9973)	SYK	Under revision.	NHL—phase I/II. (P) CLL—phase I/II. (P) AML—phase I/II. (U) ALL—phase I. (U)	Atrial fibrillation, back pain, chest pain.	ND
Erlotinib [85] (OSI-774) (Tarceva)	HER1/EGFR	Advanced or metastatic non-small cell lung cancer.	cMPD—phase I/II. (P) AML—phase II. (P)	Rash and diarrhea.	EGFR protein expression (IHC) Check CYP3A4*1B CEBPA mut.
Everolimus [86] (AFINITOR)	mTOR	Advanced renal cell carcinoma after failure of treatment with sunitinib or sorafenib.	HL—phase I/II. (U) NHL—phase I/II. (U) MM and PLCNS—phase I/II. AML—phase I/II. (U) cMPD—phase I/II. (U)	stomatitis, infections, rash, fatigue, edema, abdominal pain, fever, asthenia, cough, headache and decreased appetite.	Immunoglobulin heavy chain variable gene somatic hypermutations (Ig-V_H) Del(5q).
Galiximab [87]	CD80	Under revision	HL—phase II. (P) NHL—phase I/II/III. (U)	Abnormal liver function tests infections, low phosphate levels.	ND
Gedatolisib [41,42,43,44,45,46,47,48,49,50,51,52,53,54,55,56,57,58,59,60,61,62,63,64,65,66,67,68,69,70,71,72,73,74,75,76,77,78,79,80,81,82,83,84,85,86,87,88] (PF-05212384) (PKI-587)	PI3K-p110α/γ and mTOR	Under revision.	AML—phase II. (P) cMPD—phase II. (P)	ND	ND
Gemtuzumab [89] (Ozogamicin) (Mylotarg™)	CD33 (immunotoxin)	Acute Myeloid Leukemia CD33 positive in first relapse(patients who are 60 years of age or older).	cMPD—phase I/II. (W) ALL—phase II (W).	Chills, fever, nausea, vomiting, headache, hypotension, hypertension, hypoxia, dyspnea, hyperglycemia.	CD33 (cytofluorimetry) Del(5q)
Givinostat [90] (ITF-2357)	HDAC Inhibitor	Duchenne’s muscular dystrophy and Becker’s muscular dystrophy.	cMPD—phase II. (U) HL—phase I/II. (U) CLL—phase II. (U) MM—phase II. (P)	Gastrointestinal toxicities, cardiac toxicities, diarrhea, fatigue, nausea, thrombocytopenia, anorexia, myelosuppression.	ND
HeFi-1 [91]	CD30	Under revision	HL—phase I. (P)	ND	ND
90Y-Ibritumomab Tiuxetan [92] (Zevalin)	CD20 (immunoconjugate resulting from a stable thiourea covalent bond between the monoclonal antibody Ibritumomab and the Linker chelator tiuxetan)	Relapsed or refractory low-grade, follicular, or transformed B-cell non-Hodgkin’s lymphoma, including patients with Rituximab refractory follicular non-Hodgkin’s lymphoma.	NHL—phase II/III. (U) CLL—phase II. (U)	Neutropenia, thrombocytopenia, anemia, gastrointestinal symptoms (nausea, vomiting, abdominal pain, and diarrhea), increased cough, dyspnea, dizziness, arthralgia, anorexia, anxiety and ecchymosis. Myeloid malignancies and dysplasias.	Mutations membrane-spanning 4-domains, subfamily A, member 1 (MS4A1).
Ibrutinib [58] (Imbruvica)	Bruton’s Kinase inhibitor-BTK	Mantle cell lymphoma after at least one prior therapy Chronic lymphocytic leukemia after at least one prior therapy.	NHL—phase I/II. (U) PLCNS—phase I/II (U) cMPD—phase I. (U) ALL—phase II. (U)	Thrombocytopenia, diarrhea, neutropenia, anemia, fatigue, musculoskeletal pain, peripheral edema, upper respiratory tract infection, nausea, bruising, dyspnea, constipation, rash, abdominal pain, vomiting and decreased appetite, peripheral edema, arthralgia, stomatitis.	Chromosome 17p deletion positive (FISH) and p53 mutation screening.
Idelalisib [42] (Zydelig)	PI3K p110 δ	Relapsed CLL/SLL after at least two prior systemic therapies, Relapsed FL and B-cell NHL after at least two prior systemic therapies.	HL—phase II. (P) cMPD—phase I (P)	diarrhea, pyrexia, fatigue, nausea, cough, pneumonia, abdominal pain, chills, rash, neutropenia, hypertriglyceridemia, hyperglycemia, ALT elevations and AST elevations.	Chromosome 17p deletion positive (FISH) and p53 mutation screening.
Imatinib [23] (Gleevec)	Bcr-Abl tyrosine kinase, PDGF, SCF and c-Kit	CML ALL-Ph positive CML- PDGFR positive Aggressive systemic mastocytosis-D816V c-Kit negative Hypereosinophilic syndrome (HES) FIP1L1-PDGFRα fusion gene Recurrent and/or metastatic dermatofibrosarcoma protuberans Unresectable and/or metastatic malignant gastrointestinal stromal tumors (GIST)—Kit (CD117) positive.	cMPD—phase I/II. (U) ALL—phase II. (U) NHL—phase II. (P)	Neutropenia, thrombocytopenia, hepatotoxicity, edema, nausea, vomiting, muscle cramps, musculoskeletal pain, diarrhea, rash, fatigue and abdominal pain.	Confirm the presence of t(9;22)BCR/ABL fusion gene c-Kit-D816V mutation FIP1L1-PDGFRα.
INCB040093 [44]	PI3Kδ inhibitor	Under revision.	HL—phase I/II. (P) NHL—phase I (P) CLL—phase I.(P)	Headache, increased alkaline phosphatase, abdominal pain, pyrexia, increased ALT and AST, thrombocytopenia, neutropenia and anemia.	ND
Ipilimumab [93,94] (Yervoy)	CTLA-4	Unresectable or metastatic melanoma.	NHL—phase I/II. (U) HL—phase I. (U) MM—phase I. (U) ALL—phase I. (P) CLL—phase I. (U) CML—phase I. (U) cMPD—phase II. (U) AML—phase I. (U)	Fatigue, diarrhea, pruritus, rash, and colitis.	CTLA4 rs4553808.
Iratumumab [95] (MDX-060)	CD30	Under revision.	HL—phase I/II. (U) NHL—phase II. (U)	ND	ND
Itacitinib (INCB039110) [30]	JAK1 inhibitor	Under revision.	HL—phase I/II. (P) NHL—phase I/II. (P) cMPD—phase II. (U)	Fatigue, constipation and nausea.	ND
Ixazomib [96] (Ninlaro)	Proteasome inhibitor	Multiple myeloma after at least one prior therapy in combination with lenalidomide and dexamethasone.	NHL—phase I/II. (U) CLL—phase II. (U) AML—phase II. (U)	diarrhea, constipation, thrombocytopenia, peripheral neuropathy, nausea, peripheral edema, vomiting, and back pain.	MYC Gene Mutation.
JNJ-40346527 [97,98]	CSF-1R	Under revision.	HL—phase I/II. (P)	Edema, nausea, vomiting and headache.	ND
Lebrikizumab [99,100] (TNX-650)	IL-13	Under revision.	HL—phase I/II. (P)	Musculoskeletal, infections.	ND
Lenalidomide [101,102] (Revlimid^®^)	Immunomodulatory effects, Angiogenesis Inhibitors	Multiple myeloma, myelodysplastic syndromes.	HL—phase II (U) NHL—phase I/II/III (U). CLL—phase I/II. (U) AML—phase I/II/IV. (U) Anemia—phase III. (U) cMPD—phase II. (U)	Embryo-fetal toxicity, neutropenia, thrombocytopenia, blood clots, liver failure, skin reactions, tumor lysis syndrome, diarrhea, itching rash, tiredness. new cancers (malignancies).	Del(5q).
Lucatumumab [103] (HCD122)	CD40	Under revision.	HL—phase I/II. (U) NHL—phase I/II. (U) MM—phase II. (U) CLL—phase I. (U)	Anemia, infections, diarrhea, vomiting, pyrexia, nausea, hypotension.	ND
MDX-1401 [104]	CD30	Under revision.	HL—phase I. (P)	ND	ND
Mocetinostat [60] (MGCD0103)	HDAC Inhibitor	Myelodysplastic syndrome Diffuse large B-cell lymphoma.	HL—phase I/II. (U) NHL—phase I/II. (U) CLL—phase II. (U) cMPD—phase I/II. (U) AML—phase I/II. (U) ALL—phase I. (U) CML—phase I. (U)	Gastrointestinal toxicities, cardiac toxicities, diarrhea, fatigue, nausea, thrombocytopenia, anorexia	ND
Momelotinib [31] (CYT387)	JAK1/2	Under revision.	cMPD—phase I/II/III. (U)	thrombocytopenia, diarrhea, headache, dizziness and nausea.	JAK 1/2 mut.
Nilotinib [26] (Tasigna)	Kinase inhibitor-ABL	Chronic Myeloid Leukemia phase accelerated in adult patients resistant to or intolerant to prior therapy that included imatinib.	AML c-Kit-positive—phase I/II. (U) CML T315I positive (U)	nausea, rash, headache, fatigue, pruritus, vomiting, diarrhea, cough, constipation, arthralgia, nasopharyngitis, pyrexia, night sweats, thrombocytopenia, neutropenia and anemia.	Confirm the presence of t(9;22)BCR/ABL fusion gene c-Kit, mutation UGT1A1*28 allele homozygotes rs8175347.
Nivolumab [105] (OPDIVO)	PD-1	Melanoma and disease progression following ipilimumab and, if BRAF V600 mutation positive, a BRAF inhibitor. Metastatic squamous non-small cell lung cancer.	HL—phase I/II. (U) NHL—phase I/II. (U) AML—phase I/II. (U) MM—phase I/II. (U) CLL—phase I/II. (U) CML—phase I/II. (U) cMPD—phase I/II. (U)	Rash, fatigue, dyspnea, musculoskeletal pain, decreased appetite, cough, nausea, and constipation.	ND
Obinutuzumab [106] (GA101) (Gazyva)	CD20	Chronic lymphocytic leukemia in combination with chlorambucil.	NHL—phase I/II. (U)	Infusion reactions, neutropenia, thrombocytopenia, anemia, pyrexia, cough, and musculoskeletal disorder.	ND
Ofatumumab [107] (ARZERRA)	CD20	Chronic lymphocytic leukemia refractory to fludarabine and alemtuzumab.	HL—phase II. (U) NHL—phase I/II. (U) PLCNS—phase II. (U)	Neutropenia, pneumonia, pyrexia, cough, diarrhea, anemia, fatigue, dyspnea, rash, nausea, bronchitis and upper respiratory tract infections.	ND
Ofatumumab [108] (Arzerra) (HuMax-CD20)	CD20	Chronic lymphocytic leukemia refractory to fludarabine and alemtuzumab.	NHL—phase I/II. (U) PCN—phase II. (U)	Neutropenia, pneumonia, pyrexia, cough, diarrhea, anemia, fatigue, dyspnea, rash, nausea, bronchitis, and upper respiratory tract infections.	MS4A1.
Olaparib [109] (Lynparza)	Poly ADP Ribose Polymerase (PARP)	Ovarian cancer with BRCA mutation.	NHL—phase II. (P) MM—phase II. (P)	anemia, nausea, fatigue (including asthenia), vomiting, diarrhea, dysgeusia, dyspepsia, headache, decreased appetite, nasopharyngitis/pharyngitis, cough, arthralgia/musculoskeletal pain, myalgia, back pain, dermatitis/rash and abdominal pain/discomfort.	PARP 1 V762A.
ONO [110] (GS-4059)	BTK	Under revision.	NHL—phase I. (P) CLL—phase I/II. (U)	ND	ND
Palbociclib [111] (Ibrance)	kinase inhibitor: CDK4, CDK6	Breast cancer HER2 negative, postmenopausal women with ER positive in combination with letrozole.	MM—phase I. (P) NHL—phase II. (U) AML—phase I/II. (P) ALL—phase I/II (P)	neutropenia, leukopenia, fatigue, anemia, upper respiratory infection, nausea, stomatitis, alopecia, diarrhea, thrombocytopenia, decreased appetite, vomiting, asthenia, peripheral neuropathy, and epistaxis.	HER2 protein overexpression negative (IHC) ER+.
Panobinostat [112] (Farydak)	HDAC Inhibitor	Multiple myeloma refractory to at least 2 prior regimens, including bortezomib and an immunomodulatory agent.	cMPD—phase I/II. (U) AML—phase I/II. (U) CML—phase I/II/III. (U) ALL—phase II. (U) HL—phase I/II/III (U). NHL—phase I/II/III. (U) CLL—phase II. (U)	Gastrointestinal toxicities, cardiac toxicities, myelosuppression, hemorrhage, infections, hepatotoxicity, embryo-fetal toxicity.	JAK2V617F.
Pazopanib [113] (Votrient)	kinase inhibitor (VEGFR) PDGFR, KIT	Advance renal cell carcinoma.	MM—phase II. (U)	Increases in serum transaminase levels and bilirubin, diarrhea, hypertension, hair color changes (depigmentation), nausea, anorexia, and vomiting.	UGT1A1*28 allele homozygotes (TA)7/(TA)7 genotype.
Pembrolizumab [114] (MK-3475) (Keytruda)	PD-1	Unresectable or metastatic melanoma and disease progression following ipilimumab and, if BRAF V600 mutation positive, a BRAF inhibitor.	HL—phase I/II/III (U) NHL—phase I/II. (U) MM—phase I/II/III. (U) AML—phase II. (P) CLL—phase I/II. (U) ALL—phase II. (P) cMPD—phase I. (P)	fatigue, cough, nausea, pruritus, rash, decreased appetite, constipation, arthralgia and diarrhea.	ND
Pexidartinib [115] (PLX3397)	CSF1R, KIT and oncogenic FLT3 kinases	Under revision.	HL—phase II (P) AML—phase I/II. (U)	hair color changes, fatigue, nausea, swelling around the eyes, abnormal taste, diarrhea, vomiting, and decreased appetite, liver enzyme elevations, hyponatremia, anemia, and neutropenia.	ND
Pictilisib [45] (GDC-0941)	PI3K-p110α/δ	Under revision.	NHL—phase I. (P)	ND	ND
Polatuzumab vedotin [116] (DCDS4501A) (RG7596)	CD79b	Under revision.	NHL—phase I/II (U)	Neutropenia, peripheral sensory neuropathy, diarrhea, lung disorder, anaemia febrile.	ND
Pomalidomide [117] (Pomalyst)	Immunomodulatory effects, Angiogenesis Inhibitors	Multiple myeloma after two prior therapies including lenalidomide and bortezomib with disease progression on or within 60 days of completion of the last therapy.	NHL—phase I/II. (U) cMPD—phase I/II/III (U). AML—phase I (U)	fatigue and asthenia, neutropenia, anemia, constipation, nausea, diarrhea, dyspnea, upper-respiratory tract infections, back pain and pyrexia.	ND
Ponatinib [27] (Iclusig)	Bcr-Abl tyrosine kinase ABL, FGFR1-3, FLT3, VEGFR2	Accelerated phase, or blast phase chronic myeloid leukemia resistant or intolerant to prior tyrosine kinase inhibitor therapy, Philadelphia chromosome positive acute lymphoblastic leukemia (Ph+ALL) resistant or intolerant to prior tyrosine kinase inhibitor therapy.	AML—phase I/II (U) cMPD—phase I/II. (U)	hypertension, rash, abdominal pain, fatigue, headache, dry skin, constipation, arthralgia, nausea, and pyrexia, thrombocytopenia, anemia, neutropenia, lymphopenia, and leukopenia.	Confirm the presence of t(9;22)BCR/ABL fusion gene FLT3-mutation.
Pracinostat [118] (SB939)	HDAC Inhibitor	Under revision.	AML—phase I/II. (U) cMPD—phase I/II. (U)	Gastrointestinal toxicities, cardiac toxicities, diarrhea, fatigue, nausea, thrombocytopenia, anorexia.	ND
Resminostat [119] (4SC-201)	HDAC Inhibitor	Liver cancer.	HL—phase II. (P) NHL—phase II. (P)	Myelosuppression, gastrointestinal toxicities and fatigue.	ND
Retaspimycin [120] (IPI-504)	HSP90	Under revision.	MM—phase I. (P)	Fatigue, nausea, diarrhea, liver function test abnormalities.	ND
Rituximab [7] (Rituxan) (Mabthera)	CD20	Relapsed or refractory low-grade or follicular CD20 positive B-cell non-Hodgkin’s lymphoma.	NHL—phase I/II/III. (U) HL—phase I/II. (U) MM—phase I/II. (U) ALL-B—phase I/II. (U) CLL—phase I/II. (U) cMPD—phase II. (P)	Lymphocytopenia; HBV ractivation; severe mucocutaneous reactions (e.g., Stevens-Johnson syndrome).	MS4A1.
Romidepsin Depsipeptide [121] (Istodax))	HDAC	Cutaneous T-cell lymphoma after at least one prior systemic therapy.	NHL—phase I/II (U) MM—phase II. (U) HL—phase I/II. (P) CLL—phase I/II. (P) AML—phase II. (U) ALL—phase I/II. (U) CML—phase I. (U) cMPD—phase I/II. (U) ALL—phase I(U)	nausea, fatigue, infections, vomiting, and anorexia anemia, thrombocytopenia, ECG T-wave changes, neutropenia, and lymphopenia.	Del(5q).
Ruxolitinib [32] (Jakafi)	JAK1/2	Intermediate or high-risk myelofibrosis, including primary myelofibrosis, post-polycythemia vera myelofibrosis and post-essential thrombocythemia myelofibrosis.	HL—phase I/II. (P) NHL—phase I/II. (P) CML—phase I/II. (U) MS—phase I/II. (U) CLL—phase I/II. (U) cMPD—phase II. (U) AML—phase I/II. (U) ALL—phase II. (P)	Thrombocytopenia and anemia bruising, dizziness and headache.	JAK 1/2 mutations.
SB-743921 [122]	Kinesin spindle protein	Under revision.	HL—phase II. (P) NHL—phase II. (P)	ND	ND
SGN-30 [123]	CD30	Under revision.	HL—phase I/II. (U) NHL—phase II. (P)	Candidiasis, interstitial lung disease.	ND
Siltuximab [124] (CNTO 328) (Sylvant)	IL-6	Multicentric Castleman’s disease HIV-negative and HHV-8-negative.	NHL—phase I (P) MM—phase I. (P)	pruritus, increased weight, rash, hyperuricemia, and upper respiratory tract infection.	IL6 -174C > G rs1800795.
Sodium phenylbutyrate [125] (BUPHENYL^®^)	Urea cycle disorders HDAC Inhibitor	Urea cycle disorders. Acute promyelocytic leukemia (APL). Malignant glioma.	CLL—phase I. (U) NHL—phase I. (U) cMPD—phase I. (P) AML—phase I/II. (U) CML—phase I. (U)	Change in the frequency of breathing, lack of or irregular menstruation, lower back, side, or stomach pain, mood or mental changes muscle pain or twitching, nausea or vomiting nervousness or restlessness swelling of the feet or lower legs unpleasant taste unusual tiredness or weakness, chills, fever, joint pain, sore throat, unusual bleeding or bruising.	CEBPA mut Confirm the presence of t(15;17) PML/RAR fusion gene in APL.
Sonidegib [126] (LDE225) (Odomzo)	Hedgehog pathway inhibitor	Locally advanced basal cell carcinoma recurred following surgery or radiation therapy, or patients who are not candidates for surgery or radiation therapy. Medulloblastoma.	MM—phase II. (U) ALL—phase II. (U) AML—phase II (U) cMPD—phase I/II (U).	muscle spasms, alopecia, dysgeusia, fatigue, nausea, musculoskeletal pain, diarrhea, decreased weight, decreased appetite, myalgia, abdominal pain, headache, pain, vomiting, and pruritus.	ND
Sorafenib [51,52] (BAY43-9006) (Nexavar)	Multikinase inhibitor (BRAF and mutant BRAF, KIT, FLT-3, VEGFR-2, VEGFR-3, and PDGFR-β)	Advanced renal cell carcinoma.	NHL—phase I/II (P) HL—phase I/II. (P) AML—phase I/II/IV (U) cMPD—phase I/II. (U) CML—phase II. (U) MM and PLCNS—phase I/II (P).	Hypertension; alopecia; bleeding; rash; hand-foot syndrome; hypophosphatemia; elevated amylase and lipase levels; myelosuppression; wound-healing complications.	VEGFR, PDGFR, Kit, BRAF acquired mutation for prevention resistance CEBPA mutations, TET2 mutations, DNMT3 mutations, MLL-PTD mutations.
Sunitinib [53] (Sutent)	Multikinase inhibitor (PDGFRα, PDGFRβ VEGFR, KIT, FLT3, CSF-1R, RET)	Gastro intestinal stromal tumor after disease progression on or intolerance to imatinib mesylate. Advanced renal cell carcinoma.	cMPD—phase II (U) NHL—phase I/II (P) AML—phase I/II/IV (P). MM—phase I/II (P) ALL—phase II. (P)	Yellow of skin; hypothyroidism, adrenal function abnormalities; myelosuppression; mucositis; elevated lipase and creatinine levels; elevated liver chemistries; increased uric acid levels.	Genotype for CYP3A4. Dose reductions for CYP3A4 Poor Metabolizer (PM).
Tanespimycin [127] (KOS-953) (17-AAG)	HSP90	Under revision.	AML—phase I. (P) CML—phase I. (P) cMPD—phase I (P) NHL—phase I. (P) CLL—phase I. (P) MM—phase II/III (P). ALL—phase II. (P)	Abdominal pain, hepatotoxicity.	ND
Taselisib [46] (GDC-0032)	β isoform-sparing PI3K-p110α/δ/γ	Under revision.	NHL—phase I. (P)	ND	ND
Tazemetostat [128] (EPZ-6438)	Inhibitor of EZH2	Under revision.	NHL—phase I/II (P)	Asthenia, thrombocytopenia, nausea, constipation, anemia, dry skin, hypophosphatemia, anxiety, depression, hypertension, insomnia, peripheral edema, hepatocellular injury.	ND
Temsirolimus [129] (Torisel)	mTOR	Advanced renal cell carcinoma.	HL—phase I/Ia—(P) NHL—phase I/II. (P) cMPD—phase I/II. (P) CML—phase I/II. (P) ALL—phase I/II. (P) CLL—phase II. (P) AML—phase I/II. (P) MM and PCN—phase I/II (P).	ash, asthenia, mucositis, nausea, edema, and anorexia, anemia, hyperglycemia, hyperlipemia, hypertriglyceridemia, elevated alkaline phosphatase, elevated serum creatinine, lymphopenia, hypophosphatemia, thrombocytopenia, elevated AST, and leukopenia.	ND
TGR-1202 [47] (RP5264)	PI3K p110 δ	Under revision.	CLL—phase I/II. (P) HL—phase I/II. (P) NHL—phase I/II. (P)	Colitis and hepatic toxicity, opportunistic infections.	ND
Thalidomide [130] (Thalomid)	Immunomodulatory effects, Angiogenesis Inhibitors	Multiple myeloma in combination with bortezomib or dexamethasone.	NHL—phase I/II/I (W) HL—phase I/II. (W) CLL—phase I/II. (W)	infections, back pain, pyrexia, acute renal failure, amenorrhea, aphthous stomatitis, bile duct obstruction, carpal tunnel, diplopia, dysesthesia, dyspnea, enuresis, erythema nodosum, foot drop, galactorrhea, gynecomastia, hangover effect, hypomagnesemia, hypothyroidism, metrorrhagia, migraine, myxedema, nodular sclerosing Raynaud’s syndrome.	5p deletion.
Tositumomab [131] (Iodine-131) (Bexxar)	CD20	CD20 positive, follicular, non Hodgkin lymphoma, with and without transformation, whose disease is refractory to Rituximab and has relapsed following chemotherapy.	MM and PCN—phase I/II (U). LS—phase I/II. (U) NHL—phase I/II/III (U). HL—phase I/II. (U) CLL—phase I/II. (U)	Severe cytopenias, anaphylaxis, secondary malignancies, nausea, fatigue, infections, vomiting, anorexia, chills, Hypotension, Hypothyroidism, peripheral edema, myalgia, arthralgia.	ND
Trametinib [132] (Mekinist)	Kinase inhibitor (MEK)	Unresectable or metastatic Melanoma with BRAF V600E or V600K mutations.	MM—phase I/II. (P) AML—phase I/II. (P)	Rash, diarrhea, and lymphedema.	BRAF V600E or V600K mutations MEK protein overexpression positive (IHC).
Tremelimumab [133] (Ticilimumab) (CP-675,206)	CTLA-4	Under revision.	NHL—phase I. (U) MM—phase I (U).	Diarrhea, fatigue, hot flashes and hives.	ND
Vandetanib [50] (ZD6474) (Caprelsa)	Kinase inhibitor (EGFR-HER1/ERBB1, RET, VEGFR2)	Symptomatic or progressive medullary thyroid cancer in patients with unresectable locally advanced.	MM—phase II. (U)	Diarrhea, rash, acne, nausea, hypertension, headache, fatigue, decreased appetite and abdominal pain.	Acquired mutation for prevention resistance.
Vemurafenib [55] (Zelboraf)	Kinase inhibitor (BRAF)	Unresectable or metastatic Melanoma with BRAF V600E Mutation.	MM—phase I/II. (P) HCL—phase II. (U)	Arthralgia, rash, alopecia, fatigue, photosensitivity reaction, nausea, pruritus and skin papilloma.	BRAF V600E/K mutation positive.
Venetoclax [134] (Venclexta)	BCL-2	CLL after at least one prior therapy.	NHL—phase I. (U) MM—phase I/II. (U) cMPD—phase I. (U) AML—phase I/II. (U)	Neutropenia, diarrhea, nausea, anemia, upper respiratory tract infection, thrombocytopenia.	17p deletion.
Vismodegib [135] (Erivedge)	Hedgehog pathway inhibitor	Metastatic basal cell carcinoma, or with locally advanced basal cell carcinoma recurred following surgery or not candidates for surgery or radiation.	NHL—phase II (U) CLL—phase II. (P) cMPD—phase I/II. (U) AML—phase II (U)	muscle spasms, alopecia, dysgeusia, weight loss, fatigue, nausea, diarrhea, decreased appetite, constipation, arthralgias, vomiting, and ageusia.	ND
Vorinostat [61] (SAHA) (Zolinza)	HDAC Inhibitor	Cutaneous T-cell lymphoma, progressive, persistent or recurrent disease on or following two systemic therapies.	AML—phase I/II, (U) cMPD—phase I/II. (U) CML—phase I/II/III (U) NHL—phase I/II. (U) MM—phase I/II/III (U) HL—phase I/II. (U) CLL—phase I/II. (U) ALL—phase I/II. (U)	Diarrhea, fatigue, nausea, thrombocytopenia, anorexia, dysgeusia, creatininemia.	ND
Voxtalisib [49] (SAR245409) (XL765)	mTOR/PI3K-p110γ	Under revision.	NHL—phase I/II. (P) CLL—phase I/II. (P)	ND	ND
Ziv-aflibercept [136] (Zaltrap)	VEGF-A	Metastatic colorectal cancer resistant to or has progressed following an oxaliplatin-containing regimen, in combination with 5-fluorouracil, leucovorin, irinotecan-(FOLFIRI).	MM—phase II. (P) cMPD—phase II. (P)	leukopenia, diarrhea, neutropenia, proteinuria, AST increased, stomatitis, thrombocytopenia, ALT increased, dysphonia, serum creatinine increased.	ND

^$^ The molecules should be either in preclinical stage (P), in clinical use (U) and/or withdrawn/obsolete (W). Abbreviations. ALL: Acute lymphoblastic leukemia; AML: Acute myeloid leukemia; CML: Chronic myeloid leukemia; CLL/SLL: Chronic lymphocytic leukemia/small lymphocytic leukemia; DLBCL: Diffuse large B cell lymphoma; HL: Hodgkin lymphoma; FL: Follicular lymphoma; HCL: Hairy cell leukemia; HG-BCL: High grade B-cell lymphoma; IGHV: Immunoglobulin G heavy variable chain; iNHL: indolent NHL; IV: intravenous; MCL: Mantle cell lymphoma; MM: Myeloma multiple; MDP: Myeloproliferative disorders; MZL: Marginal zone lymphoma; PLCNS: Primary lymphoma of central nervous system; ND: Not documented; pts: patient. Note: the present table could be not exhaustive of all current agents used for onco-hematology.

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
