# Peer review of "Overview of Current Targeted Anti-Cancer Drugs for Therapy in Onco-Hematology"

_medicina, 2019, doi:10.3390/medicina55080414_

Round 1
Reviewer 1 Report
The review is well written and properly describes the pharmacological advancements in the field of onco-haematological drugs. It is my opininon the review should be published as it is.
Author Response
R. I want to thank the reviewers for their positive evaluation of the manuscript.
Reviewer 2 Report
Crisci and coauthors have reviewed in the paper the targeted agents for hematological cancers currently undergoing I and II phase of clinical trials (Based on ClinicalTrials, Febuary 2017). In particular the review describes the monoclonal antibodies and small molecule inhibitors, i.e.: HDAC, proteosome, Kinase-based Signalling and Bruton’s Tyrosin Kinase inhibitors. Although the ClinicalTrial database is readily available for readers, the authors gathered the particular data and present as a summary with addition of their opinion on the role of targeted therapy. I found it valuable and interesting not only for clinicists but also for a broad spectrum of scientists.
I recommend this article for publication, however I recommend the language editing due to a number of editorial and language blunders, e.g.:.
Finally, low (a few?) studies published to date shown high variability and unpredictability about patient adherence to oral cancer treatment regimens
Drugs that block growth signaling by bind (binding?) to receptors on the cell surface
The article bases on the data collected February 2017 and it should be updated.
Please clarify, why the study presented in the Table 1 concerns also other types of cancer (e.g. melanoma, breast cancer). Shouldn’t these results (molecules) be removed?
The authors are also advised to clarify if the described in the text molecules are in – use, or are at the stage of clinical trials (described in the Table 1) or at the preclinical stage.
Chapter 3. Pharmacogenomic Test is to general. Please add a few examples supported by references.
The references need to be unified.
According to the article title the data concerns only onco-hematology. Please adjust the conclusion to this narrow oncology division or adjust the title to be more general.
Page 2 line 91, please add the examples of traditional drugs and its mechanism of action.
Page 3 line 110 what authors mean as a toxins
Table1. Heading should start with a capital letter
Author Response
R. I want to thank the reviewers for their valuable suggestions
I recommend the language editing due to a number of editorial and language blunders, e.g.:.
The early evaluation of the English language has been made by English native spoken.
Finally, low (a few?) studies published to date shown high variability and unpredictability about patient adherence to oral cancer treatment regimens
R. It has been edited
Drugs that block growth signaling by bind (binding?) to receptors on the cell surface
R. It has been edited.
The article bases on the data collected February 2017 and it should be updated.
R. It has been updated to may 2019
Please clarify, why the study presented in the Table 1 concerns also other types of cancer (e.g. melanoma, breast cancer). Shouldn’t these results (molecules) be removed?
R. Since, several molecules have been FDA-approved for non-oncohematology malignancy, we cited them on table 1 when trials in onco-hematology were performed.
The authors are also advised to clarify if the described in the text molecules are in – use, or are at the stage of clinical trials (described in the Table 1) or at the preclinical stage.
R. It has been specified $The molecules should be either in preclinical stage (P), in clinical use (U) and/or withdrawn/obsolete (W)
Chapter 3. Pharmacogenomic Test is to general. Please add a few examples supported by references.
A specific example of the therapy directly dependent from pharmacogenomics test was cited “The first example was Dasatinib and Nilotinib designed to by-pass the acquired mutation T315I in Abl gene in LMC patients”
The references need to be unified.
R. It has been unified as “instruction for authors” rules.
According to the article title the data concerns only onco-hematology. Please adjust the conclusion to this narrow oncology division or adjust the title to be more general.
Several sentences have been updated concerning onco-hematology.
Page 2 line 91, please add the examples of traditional drugs and its mechanism of action.
R: it was added “………like as cytoskeleton inhibitors (i.e tecans, vincristine etc), analogues of nucletides (i.e 5-Fluorouracil, Gemcitabine etc). So each normal cell in mitotic stage……”
Page 3 line 110 what authors mean as a toxins
In the draft was added “(viral molecules to interfere with cell vitality)”. An example of this kind of molecule is brentuximab Vedotin.
Table1. Heading should start with a capital letter
R. It has been edited.
Reviewer 3 Report
Authors present a very detailed review of current targeted anti-cancer drugs used in oncology. Different modalities of targeted therapy have been presented in short, concise and understandable manner for broad audience.
Authors have put basic information which healthcare professionals need to know on targeted oncological therapy. In the chapter biological mechanism of action of targeting agents it would be good to show mechanism of action of one of the agents on the example (for example, mechanism of action of one concrete targeted drug which is used as a standard therapy in a certain disease). In the chapter on kinase-based signaling inhibitors it would be interesting to read more on promising therapy in malignant melanoma, especially with encouraging reports on effectiveness of BRAFtovi and MEKtovi therapy. Pharmacogenomic testing should be explained on an example (specific disease in which the testing is used, specification of the results and the consequence - use of a certain medication). Table provides a very detailed insight into targeted therapy. It would be interesting to add few sentences on possibilities of use of targeted therapy in the tumors of central nervous system.
Author Response
R. I want to thank the reviewers for their valuable suggestions
In the chapter on kinase-based signaling inhibitors it would be interesting to read more on promising therapy in malignant melanoma, especially with encouraging reports on effectiveness of BRAFtovi and MEKtovi therapy. Pharmacogenomic testing should be explained on an example (specific disease in which the testing is used, specification of the results and the consequence - use of a certain medication). Table provides a very detailed insight into targeted therapy. It would be interesting to add few sentences on possibilities of use of targeted therapy in the tumors of central nervous system.
It was added a specific sentence in the Pharmacogenomic test heading: “Recently, to reduce pharmaceutical expenditure, a combination of 2 SMIs blocking 2 mutated genes (BRAF and MET) are performed in the unique formulation (encorafenib+binimatenib) for melanoma therapy. So, the route is also drawn for the onco-hematology”.
In addition, a specific example of the therapy directly dependent from pharmacogenomics test was cited “The first example was Dasatinib and Nilotinib designed to by-pass the acquired mutation T315I in Abl gene in LMC patients”
A brief consideration of CNS Lymphoma was added: “Unlikely, no upgrading was seen for Primary Lymphoma Central Nervous System (PLCNS) in term of targeted therapy. The use of SMIs in PCNS and as preventing metastasis has failed. [151]. Encouraging evidence supporting the addition rituximab to classical alkylating agents is growing, and a recent randomized trial (MATRix regimen) demonstrated a significantly better OS [152]”